# Position: The Future of Bayesian Prediction Is Prior-Fitted

**Samuel Müller** [1 2] **Arik Reuter** [3] **Noah Hollmann** [4] **David Rügamer** [3 5] **Frank Hutter** [6 1 4]

## Abstract

Training neural networks on randomly generated artificial datasets yields Bayesian models that capture the prior defined by the dataset-generating distribution. Prior-data Fitted Networks (PFNs) are a class of methods designed to leverage this insight. In an era of rapidly increasing computational resources for pre-training and a near stagnation in the generation of new real-world data in many applications, PFNs are poised to play a more important role across a wide range of applications. They enable the efficient allocation of pre-training compute to low-data scenarios. Originally applied to small Bayesian modeling tasks, the field of PFNs has significantly expanded to address more complex domains and larger datasets. This position paper argues that PFNs and other amortized inference approaches represent the future of Bayesian inference, leveraging amortized learning to tackle data-scarce problems. We thus believe they are a fruitful area of research. In this position paper, we explore their potential and directions to address their current limitations.

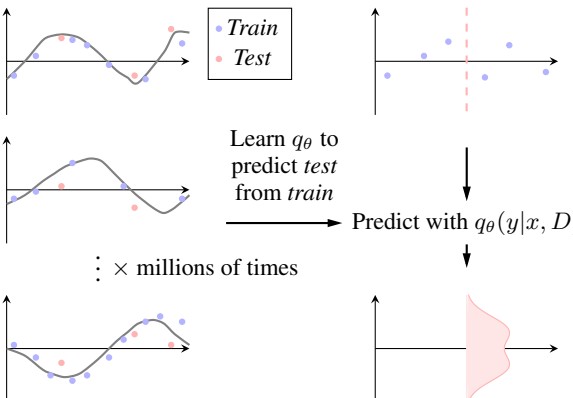

(a) Example of prior-fitting and inference on a 1D prior

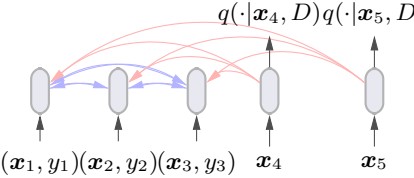

(b) Original architecture (Müller et al., 2022)

*Figure 1.* (a) The PFN learns to approximate the Bayesian prediction offline by training on datasets sampled from the prior and transfers to real-world data. (b) In a typical PFN architecture training samples $(x_i, y_i)$ can attend only to each other; test positions ($x_4$ and $x_5$) attend only to the training positions.

## 1. Introduction

The computing costs to pre-train neural networks continue to rapidly decrease, especially through improvements in GPU manufacturing (Epoch AI, 2024a), as well as hardware-utilization, optimization, and neural network architectures (Ho et al., 2024).

However, the amount of available real-world data does not scale at the same pace as compute in various domains. In this work, we argue that training on vast quantities of synthetic data is ideally suited to utilize the rapidly increasing amount

[1]University of Freiburg, Freiburg, Germany [2]Meta, New York (work done at University of Freiburg) [3]LMU Munich, Munich, Germany [4]Prior Labs [5]Munich Center for Machine Learning (MCML), Munich, Germany [6]ELLIS Institute Tübingen, Tübingen, Germany. Correspondence to: Samuel Müller <sammuller@meta.com>.

*Proceedings of the 42$^{nd}$ International Conference on Machine Learning*, Vancouver, Canada. PMLR 267, 2025. Copyright 2025 by the author(s).

of available computational resources for neural network pre-training in these areas.

Initially, the most prominent application of synthetic data involved modest modifications of real-world datasets through data augmentation techniques. These techniques have, for instance, become indispensable in image classification tasks with limited data (Deng et al., 2009; Cubuk et al., 2019; Müller & Hutter, 2021). Since then, the use of completely artificial data, defined by human-designed algorithms, has expanded to various domains, including tabular supervised learning (Hollmann et al., 2023; 2025), symbolic regression (d'Ascoli et al., 2022; Kamienny et al., 2022), geometric reasoning (Trinh et al., 2024), and causal discovery (Lorch et al., 2022). This widespread adoption underscores the versatility and effectiveness of pre-training on synthetic data to leverage compute in order to improve performance.

In this paper, we advocate for a general approach for exploiting synthetic data in pre-training: Prior-Data Fitted Networks (PFNs, Müller et al., 2022). PFNs are a Bayesian prediction method. They directly approximate the posterior predictive distribution (PPD), diverging from traditional Bayesian methods that depend on explicit likelihood functions, intricate sampling or variational inference techniques. Instead, PFNs utilize neural networks trained through supervised learning to perform Bayesian prediction directly via in-context learning (ICL). PFNs are thus distinct from other large-scale neural networks: they are solely trained on artificial data, and only conditioned on real-world data. PFNs were shown to be over $200\times$ faster than previous methods at Bayesian prediction on small scale data (Müller et al., 2022) and over $10\,000\times$ faster for Bayesian learning curve extrapolation (Adriaensen et al., 2023).

PFNs are neural networks, directly optimized to perform Bayesian predictions in context: Given a set of observed data points and a query point, they predict the posterior predictive distribution for that query, as exemplified in Figure 1a. They learn to do this by training on samples from an artificial prior distribution over datasets (Figure 1). While pre-training the PFN for a given prior can be expensive, the application to a new dataset is fast: it corresponds only to a forward pass for the current PFN architectures (Müller et al., 2022; Hollmann et al., 2025).

PFNs dramatically expand the space of possible priors that can be practically used in Bayesian inference. Traditional methods typically require priors with, for example, tractable likelihood functions. In contrast, PFNs only require the ability to sample from the prior, which can be specified implicitly by a generative process or simulation. This enables Bayesian modeling based on rich, domain-specific modeling assumptions that would be intractable with conventional approaches.

PFNs are already being applied to domains as diverse as time series forecasting (Verdenius et al., 2024; Hoo et al., 2025; Bhethanabhotla et al., 2024), outlier detection (Shen et al., 2024), Bayesian optimization (Müller et al., 2023c; Rakotoarison et al., 2024), tabular regression (Hollmann et al., 2025) and classification (Hollmann et al., 2023; Müller et al., 2023a; Xu et al., 2024), learning curve extrapolation (Adriaensen et al., 2023), biology applications (Scheuer et al., 2024; Ubbens et al., 2023; Czolbe & Dalca, 2023).

> **Position**
>
> **This paper argues that Prior-Data Fitted Networks (PFNs) are a fruitful area of research, as they will dominate most applications of Bayesian prediction and create new ones in the future.**

A central reason for this belief is the existing trend in the domain where PFNs were first applied: tabular data. The poster child of PFNs is TabPFN (Hollmann et al., 2023; 2025), a breakthrough in tabular machine learning, as the first deep learning model that consistently outperforms classic methods like XGBoost (Chen & Guestrin, 2016) on small tabular datasets with up to $10\,000$ examples, yielding better performance in 5 seconds than any baseline reached with 4 hours of tuning (Hollmann et al., 2025). Moreover, PFNs have enabled various new applications of Bayesian methods for predictions in domains as diverse as RNA folding times (Scheuer et al., 2024), computer-chip latencies (Carstensen et al., 2024), and metagenomics data (Perciballi et al., 2024). On a fundamental level, PFNs are uniquely positioned to leverage abundant pre-training compute in data-scarce environments. However, realizing their full potential requires addressing key open challenges, which we outline to guide future research.

## 2. Background

In this section, we detail PFNs and place them into context.

### 2.1. Prior-data Fitted Networks

Prior-data Fitted Networks (PFNs) are pre-trained (*prior-fitted*) to approximate the posterior predictive distribution (PPD) and thus perform Bayesian predictions for a particular prior. They directly approximate the PPD without instantiating a posterior over latents unlike most generic Bayesian prediction methods. During pre-training, we assume that there is a sampling scheme for the prior, such that we can sample datasets of inputs and outputs: $D \sim p(D)$. This requirement is easily satisfied for a large class of priors. In most cases discussed in this paper, we model the prior over datasets using a prior $p(\xi)$ over latent variables $\xi$. These variables can take various forms, such as graphs, finite and infinite vectors, among others, as well as mixtures of these (Hollmann et al., 2023; Müller et al., 2022). We then sample the dataset as $D \sim p(D|\xi)$, based on a sampled $\xi$. We repeatedly sample synthetic datasets $D = \{(\boldsymbol{x}_i, y_i)\}_{i \in \{1,\dots,n\}} \sim p(D)$ and optimize the PFN's parameters $\theta$ to make predictions for $(\boldsymbol{x}_{test}, y_{test}) \in D$, conditioned on the rest of the dataset $D_{train} = D \setminus \{(\boldsymbol{x}_{test}, y_{test})\}$. The PFN $q_\theta$ can be considered an approximation to the PPD. It accepts a training set as well as a test input and returns a distribution over outcomes for the test input. The loss in PFN pre-training is the cross-entropy on the held-out examples

$$\ell = \mathop{\mathbb{E}}_{\substack{\{(\boldsymbol{x}_{test}, y_{test})\} \cup D_{train} \\ \sim p(D)}} [-\log q_\theta(y_{test}|\boldsymbol{x}_{test}, D_{train})].$$

Minimizing this loss approximates the true posterior predictive distribution (PPD) (Müller et al., 2022; Goodfellow et al., 2016), as it is the KL-divergence to the true PPD

across datasets in the prior

$$\ell = \mathop{\mathbb{E}}_{\substack{x, D_{train} \\ \sim p(D)}} \left[ \text{KL}(p(\cdot|x, D), q_\theta(\cdot|x, D)) \right] + C.$$

This loss can incorporate Bayesian models with latents easily by sampling the latents $\xi \sim p(\xi)$ first and using the data likelihood to sample datasets based on them $p(D) \sim p(D|\xi)$. Crucially, this *prior-fitting* phase is performed only once for a given prior $p(D)$ as part of algorithm development. It can be viewed as an initial phase to learn how to learn on new data, similar to meta-learning methods (Finn et al., 2017; Santoro et al., 2016) but trained on synthetic instead of real-world data. For technical details of the training procedure for the experiments in this position paper, we refer to Appendix A.

## 2.2. Examples of PFN Priors

The crucial aspect of PFNs is the synthetic data generation process, which implicitly defines the prior $p(D)$. Below, we detail a few illustrative examples.

**Bayesian Neural Network Priors**  Multiple works (Müller et al., 2022; 2023c; Hollmann et al., 2023) explored a Bayesian neural network (BNN) prior, using a multi-layer perceptron (MLP) architecture. The prior here is defined over the BNN's weights. The PFN trained on this prior will approximate the predictions of a BNN, an MLP with uncertainty over it's weight and a prior $\mathcal{N}(0, \sigma^2)$ over its weights. Synthetic datasets are generated by:

1. Sampling the MLP weights $\xi$ i.i.d. from a prior distribution, e.g., $\xi_{jk} \sim \mathcal{N}(0, \sigma^2)$.
2. Sampling input features $\boldsymbol{x}_i$ i.i.d. from a simple distribution, e.g., $U([0, 1]^d)$.
3. Generating target values $y_i$ by passing $\boldsymbol{x}_i$ through the sampled neural network $f_\phi(\boldsymbol{x}_i)$.

This prior is very broad, encompassing all functions representable by the chosen MLP architecture and weight prior.

**Gaussian Process Priors**  Müller et al. (2022) and Müller et al. (2023c) utilized Gaussian Process (GP) priors to approximate GPs that are Bayesian over their hyperparameters with PFNs, as they are a popular choice for surrogate modeling in BO. This can be particularly useful for Bayesian optimization (BO), as shown by Müller et al. (2023c). As it is fully Bayesian over hyperparameters, it involves a "meta" prior over the GP hyperparameters (e.g., length scales, kernel types, output scale). The generation of one synthetic dataset $D = \{(\boldsymbol{x}_i, y_i)\}$ during pre-training proceeds as follows:

1. Sample GP hyperparameters $\xi$ from their respective hyper-prior distributions (e.g., uniform over a range).
2. Sample input features $\boldsymbol{x}_i \sim U([0, 1]^d)$.

3. Sample target values $y_i$ from the GP defined by the sampled hyperparameters $\xi$: $\boldsymbol{y} \sim \mathcal{N}(\boldsymbol{0}, K_\xi)$, where $K_\xi$ is the kernel matrix.

This process generates diverse datasets, reflecting a wide range of possible underlying functions one might encounter in, e.g., BO.

**The TabPFN Prior**  While the previous two examples tread close to priors used with other Bayesian methods, TabPFN (Hollmann et al., 2023; 2025) employs a highly sophisticated prior tailored to PFNs. This prior was build to perform well for supervised learning on tabular data. It is a prior over structural causal models (SCM), which are computation graphs with linear connections, activation functions in each node, noise in each node. The features and target are the values read off at random nodes in the graph. This is a prior unthinkable to approximate the posterior for with traditional methods, as the latent involves a complex graph and a large set of possible activation functions. Key characteristics include:

**A Learning Curve Prior**  Adriaensen et al. (2023) designed a prior specifically to mimic learning curves observed in machine learning training processes, capturing typical shapes like power laws or sigmoidal functions.

**A Time Series Prior**  Dooley et al. (2023) developed a prior for time-series forecasting that includes common temporal patterns, such as seasonality (e.g., weekly or yearly cycles) and trends.

These examples show how domain knowledge can be encoded into the data generation process to create specialized PFNs, but also broad models applicable to large domains. The prior is implicitly defined by the algorithm that generates synthetic datasets. This declarative approach allows practitioners to tailor the PFN's inductive biases to the problem domain by controlling the characteristics of the data it learns from during pre-training.

## 2.3. Relationship of PFNs to Traditional Bayesian Prediction Methods

Traditionally, Bayesian inference and prediction are conducted using methods that operate on a per-dataset basis and do not amortize across datasets for a particular prior. Prominent methods for supervised learning are the following:

i) Markov Chain Monte Carlo (MCMC, Neal, 1996; Andrieu et al., 2003; Welling & Teh, 2011) methods, such as NUTS (Hoffman et al., 2014), provide accurate but sometimes very slow approximations of the posterior; and

ii) Variational Inference (VI, Jordan et al., 1999; Wainwright & Jordan, 2008; Hoffman et al., 2013) methods approximate the posterior using a tractable distribu-

tion, such as a factorized normal distribution, which inherently limits the exactness of such methods.

iii) Finally, Gaussian processes (GPs, Rasmussen & Williams, 2006) are a method to allow predicting with infinite-dimensional latents, namely all smooth functions.

We believe that PFNs hold great promise to supersede these methods for (supervised) prediction tasks, as they are not only the only method that allows defining the prior declaratively as outlined in Section 3, but additionally the only method that can do all of the following:

**Simplicity of implementation** MCMC and VI methods can be hard to implement correctly. In contrast, PFNs simply execute a forward pass on a relatively standard neural architecture, a highly standardized procedure that could even be compiled to ONNX (developers, 2021).

**Handle complex latent distributions** PFNs can handle probabilistic models with very complex latent distributions, as the latents are never modeled explicitly. This contrasts with MCMC, which can converge very slowly for large latent dimensions, and VI, for which one needs to explicitly parameterize the latent distribution, including interactions between latent variables. For example, while a Bayesian treatment of neural networks typically only aims for a posterior distribution of the weights of a fixed architecture, PFNs trivially allow defining a prior over architectures as well. In MCMC, this would require advanced techniques like reversible jumps (Green, 1995), and in VI parameterizing a variational posterior over different architectures can incur various problems (Rudner et al., 2022).

**Approximate a large class of priors** Unlike GPs, PFNs, as well as MCMC and VI, can be used for a large class of priors. The prerequisites for PFNs are slightly different; they require the ability to sample datasets from the prior, whereas MCMC and VI typically necessitate the ability to compute both the density of the data $p(D|\xi)$ and the prior probability $p(\xi)$, where $\xi$ denotes the latent variables.

**Return Predictions without Sampling** Unlike VI and MCMC, PFNs and GPs model the predictive distribution directly and thus do not need to sample latents from the posterior to approximate it (Blundell et al., 2015).

We further think that PFNs hold great promise to supersede other methods for (supervised) prediction tasks, as they can effectively utilize neural network training compute, which we believe to continue to scale exponentially and faster than specialized inference compute due to the intense focus on it in the industry (Epoch AI, 2024b;a; Liu et al., 2024).

Further, we believe that the amortization of compute across tasks is still underexplored in most applications of Bayesian inference: many areas could amortize compute to perform a lot of different Bayesian predictions.

## 2.4. Relationship of PFNs to Other Amortized Deep Learning Methods

While this paper focuses on PFNs, which work on datasets and make predictions for test samples, PFNs are just part of a larger development of using deep learning to amortize probabilistic models on simulated data. Other prominent candidates from this field include the following approaches.

**Amortized Simulation-Based Inference** Simulation-based inference (SBI; Cranmer et al., 2020) performs Bayesian inference for the latent parameter $\xi$ that determines the behavior of scientific simulations. While simulators allow sampling from the joint distribution $p(\xi, \boldsymbol{x})$ of the latent $\xi$ and data $\boldsymbol{x}$, SBI methods aim to obtain insight into the typically multivariate posterior $p(\xi|\boldsymbol{x}^*)$, where typically one specific dataset $\boldsymbol{x}^*$ is considered. Recently, amortized neural posterior estimation (NPE) methods have substantially gained importance (Papamakarios & Murray, 2016; Greenberg et al., 2019), where the goal is to approximate $p(\xi|\boldsymbol{x})$ with a model $q_\theta(\xi|\boldsymbol{x})$ for arbitrary $\boldsymbol{x} \sim p(\boldsymbol{x})$. In NPE, the objective function is given by

$$\mathbb{E}_{(\xi,\boldsymbol{x})\sim p(\xi,\boldsymbol{x})} \left[-\log q_\theta(\xi|\boldsymbol{x})\right]. \qquad (1)$$

This is very similar to the PFN objective, but PFNs model the simple posterior predictive instead of the posterior of the latent $\xi$. Besides using NPE with normalizing flows (Papamakarios, 2019; Wirnsberger et al., 2022), current approaches utilize diffusion and flow matching (Gloeckler et al., 2024; Wildberger et al., 2024) to model the latent's posterior. Unlike PFNs, SBI methods typically target high-dimensional complex posterior distributions of parameters within scientific simulations and are often closely motivated by specific applications, for example in physics (Gebhard et al., 2025), or neuroscience (Lueckmann et al., 2019; Manzano-Patrón et al., 2024).

**Causal Discovery** Beyond posterior distributions arising from scientific simulations, amortized inference has been applied to causal structure learning (Lorch et al., 2022), where an objective analogous to Equation 1 is used to learn the structure of causal graphs based on synthetically generated pairs of causal graphs and observational or interventional data.

**Amortized Symbolic Regression and Reasoning** A different field that models latents by training on randomly generated data, like the methods above, are neural symbolic regression approaches. Here, the latent is a formula describing an input-output relation (Kamienny et al., 2022) or a recurrence (d'Ascoli et al., 2022). The focus is thus less on approximating the distribution, but rather on finding a single formula that models the data at hand. Symbolic

regression methods typically model the multi-dimensional latent distribution using a transformer encoder (Kamienny et al., 2022; d'Ascoli et al., 2022). Symbolic regression is of utmost interest in many scientific applications. However, in practical applications where the prediction quality is the main concern, direct modeling of the simple prediction distribution in PFNs is an advantage.

**Neural Processes**   Neural processes (NPs, Garnelo et al., 2018a;b) are neural networks that function as stochastic processes, materializing in a permutation invariance in both training and testing samples. The neural network architectures used for PFNs, as exemplified in Figure 1b, typically also meet the criteria of conditioned stochastic processes, thus of NPs. Additionally, PFNs have, similar to Conditional NPs (Garnelo et al., 2018a) a fully factorized output distribution treating each output separately, see Figure 1b. The architectural similarity is particularly evident when comparing the original PFN architecture (Müller et al., 2022) to the port of the PFN architecture to NP applications, the Transformer NP (Nguyen & Grover, 2022).

While architecturally related to neural processes, PFNs are a method to perform amortized inference with a particular focus on designing data generation processes. Rather than learning from fixed real-world datasets, like NPs, PFNs learn-to-learn and thus approximate Bayesian prediction from synthetic data sampled from a prior.

## 3. Scope and Limitations of PFNs

In this section we outline what PFNs already offer and what their current limitations are.

**Declarative programming of prediction algorithms**
Unlike traditional prediction algorithms, such as random forests (Breiman, 2001), that are defined imperatively, PFNs enable the declarative definition of algorithms by specifying prior distributions $p(D)$ over datasets. This allows the direct encoding of assumptions (e.g., Gamma-distributed noise, linear relationships, imbalance or missing values) by including corresponding examples during pre-training. This declarative approach simplifies domain-specific model engineering by allowing practitioners to directly incorporate relevant dataset characteristics into the prior.

**Unlimited data**   Real-world data tends to run out in many domains compared to compute, which scales exponentially (Epoch AI, 2024b;a). PFNs are thus a timely approach to use more compute to improve performance without relying on more real-world data. PFNs currently already excel on small data problems, small Bayesian optimization problems (Müller et al., 2023c) and tabular supervised learning (Hollmann et al., 2025), but one can expect that the area where these models excels grows as the divide between compute and real-world data availability widens.

**No Possibility for Data Leakage**   As PFNs are pre-trained on synthetic data, there is no possibility for data leakage to occur. This can be of high relevance for domains, where both PFNs and foundation models trained on real-world data exist. We can see this advantage already playing out for time series forecasting with foundation models, where dataset leakage is a big problem, and a PFN is among the top-performing models (Hoo et al., 2025).

**Current Limitations**   Current PFNs also have downsides for use in new applications, compared to traditional methods. Here, we list currently known shortcomings and references to ideas for how to address them.

1. PFNs can be less interpretable compared to traditional methods, as they hide the latent from the user. See Sections 4.4 and 5.4 for approaches to alleviate this problem.

2. The support set of datasets and their generating distributions is typically limited and less well defined for PFNs compared to other Bayesian methods. Therefore, it is less clear what data they work well on. This problem can be addressed in different ways, detailed in Section 5.

3. Currently, PFNs work best for smaller datasets and are commonly outperformed on large datasets. While there are potentially fundamental limitations in learning from large-scale data in context, we believe most of the current reasons are efficiency- and compute-related, which we can make progress on as described in Section 6.1.

4. PFNs are not well positioned for tasks where fast inference of new test datapoints is crucial. Their inference times tend to be much slower compared to traditional methods in tabular settings, for example (Hollmann et al., 2025). This is mostly an engineering challenge, though, and not fundamental as we outline in Section 6.3.

5. Current PFN architectures exhibit specific modeling limitations, which we address in Section 6.5.

In the following sections, we will explore exciting research opportunities with PFNs.

## 4. PFN Extensions

### 4.1. Incorporating Extra Inputs

PFNs can be extended to accept additional contextual inputs beyond the dataset itself, enabling adaptive behavior based on user preferences and domain knowledge. The PFN then learns to approximate the PPD $p(y|\boldsymbol{x}, D, \rho)$ conditioned on this extra information $\rho$. Recent work has already demonstrated the practical value of this approach: Müller et al. (2023c) incorporated user beliefs about optima in black-box optimization, Helli et al. (2024) used a distribution indicator to handle distribution shifts, and FairPFN (Robertson et al., 2024) achieved approximate counterfactual fairness (Kusner et al., 2017) through protected feature inputs.

**Algorithm 1** Latent Prior and Sampling Functions

```
1: function latent_prior()
2:    w ~ N(0, 𝕀)
3:    return w
4: end function
5: function sample_single_example(w)
6:    x ~ N(0, 𝕀)
7:    y ~ wᵀx + N(0, 0.1 · 𝕀)
8:    return (x, y)
9: end function
```

This flexibility opens exciting possibilities for user-friendly prediction systems. Promising input types include noise levels, functional constraints (e.g., homogeneity), extrema characteristics, complexity and smoothness parameters, and feature-output relationship specifications.

### 4.2. An In-Context Interpreter

Building upon the additional user inputs presented above, we propose to upgrade PFNs to be in-context interpreters for probabilistic model definitions. Here, users can specify their knowledge through code that implicitly defines a prior over datasets $p(D)$. The PFN models the PPD $p(y|x, D, \rho)$ conditioned on a program $\rho$ specified at prediction time. This moves PFNs beyond fixed priors to become in-context interpreters for probabilistic programming languages (PPLs). As a first step, this could be a PPL for supervised learning with a latent-based prior, as exemplified in Algorithm 1. A *latent_prior*() function specifies how global latents are sampled, and a *sample_single_example*() function describes how to use the dataset-level latent to generate one sample. In Algorithm 1, we use this format to define a linear regression prior. The training process would involve sampling a large set of random programs $\rho$ along with datasets sampled with them and amortize over different programs.

Current PFNs simplify the definition of prediction algorithms by specifying them as data generation mechanisms and then training on these mechanisms. The proposed approach simplifies this further by removing the training step, decreasing turnaround times in algorithm development dramatically. This might, for example, allow the definition of a new TabPFN (Hollmann et al., 2025) in a single prompt.

### 4.3. Bayesian Optimization via Reinforcement Learning

Bayesian Optimization is a blackbox optimization technique with the goal to find a maximal point of an unknown function $f$ after $K$ queries $x_k$, $k < K$. The goal commonly is defined as maximizing $g = \sum_{0 < k \le K} \mathbb{E}[f(x_k)]$. The key idea of Bayesian optimization is to assume the function was sampled from a known prior $p(f)$. Even under this assumption, though, approximating the optimal next query for queries more than a few steps away from $K$ is infeasible

with current methods (Garnett, 2023).

The first work on PFNs for Bayesian optimization (Müller et al., 2023c) attempted to approximate Bayesian optimization by stacking lookahead models. However, this approach involves training numerous models, each relying on the output of the preceding model, which tends to degrade performance with an increasing number of steps. We believe that there is much to gain by taking inspiration from the advancements in "reasoning" language models (Kimi et al., 2025; DeepSeek-AI et al., 2025) and trying to use simple reinforcement learning (Sutton & Barto, 2018) to teach the model to trade-off exploration and exploitation on prior functions using the goal $g$ summed only over future steps as reward, after an initial classical PFN training phase.

### 4.4. Latent Prediction

PFNs, unlike other amortized inference methods, do not model the latent. This is an advantage for most prediction settings, as predictive distributions are commonly one-dimensional, while the latent is high-dimensional. The drawback of this direct approach is that one can only access the model's internal posterior representation via its predictions.

While PFNs were already shown to effectively learn posterior distributions over latent variables (Reuter et al., 2025), there might be positive transfer between both tasks. And users of prediction models could benefit from understanding the model's reasoning better through latents. Most priors will have a complex latent, but an auto-regressive modeling approach, as proposed by d'Ascoli et al. (2022) and Kamienny et al. (2022), can still model that. For modeling graph-based latents, like in TabPFN, or real-valued vector-based latents, one can consider the methods proposed by Lorch et al. (2022) and Reuter et al. (2025), respectively.

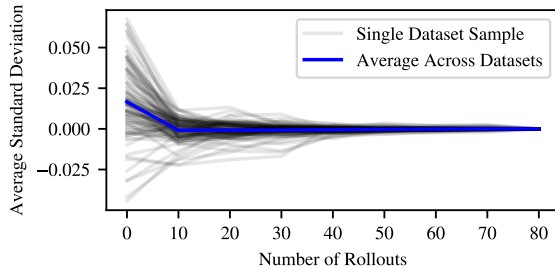

*Figure 2.* We show the average standard deviation of 100 datasets sampled from our prior, each normalized by its final standard deviation. We can see that the standard deviations tend to shrink in the first few steps of each roll-out. This should not be the case for a Bayesian predictor, but it stabilizes after some steps, as expected. Further, the deviations in standard deviation are small in absolute terms, compared to an average standard deviation of 0.21.

# 5. Understanding PFNs

It was previously shown that PFNs and similar ICL setups can perform Bayesian prediction on many priors (Müller et al., 2022; Bai et al., 2024). Still, the behavior of PFNs is not clear in many circumstances, and understanding their behavior in general is crucial for researchers to trust them.

## 5.1. Martingale Perspective

The most relevant question in understanding PFNs, in our opinion, is which features of true Bayesian predictions they fulfill and which they don't in which circumstances. Recently, the Martingale property (Falck et al., 2024) was proposed to do this for in-context learners. The Martingale property states that the models' posterior prediction distribution approximation $q_\theta(y|x, D)$ is roughly equal to the approximation after sampling from their output distribution $n$ times sequentially, where we condition on the previous samples:

$$q'_\theta(y|x, D) \approx \mathbb{E}_{x_i \sim p(x);\ y_i \sim p(y_i|x_i, D \cup \{(x_j, y_j)\}_{0<j<i})}$$
$$[q_\theta(y|x, D \cup \{(x_j, y_j)\}_{0<j<n})].$$

Both should approximate the PPD, so both should be roughly equal. Falck et al. (2024) found that language models do not fulfill this property well. We started an exploration on this in Figure 2, where we can see that models do on average tend to slightly decrease their average standard deviation for the first few steps in violation of the Martingale property, but stabilize after that, fulfilling the Martingale property. We show some example trajectories in the Appendix in Figure 4. While this is only a very small-scale experiment, we include it to show the kind of directions one might take to better understand PFNs.

## 5.2. Limit Behavior

A crucial question is under what conditions PFNs can learn from additional data, particularly when it is outside their prior. Nagler (2023) published an initial paper examining some PFN architectures at scale, revealing potential for PFN architectures to surpass current large-scale approaches, warranting further exploration.

## 5.3. Impact of Prior Density

PFNs approximate Bayesian prediction, which fundamentally requires the dataset to have support in the prior. While PFNs generally inherit this dependency, they sometimes can, similar to other neural networks, generalize beyond the prior's support (Hollmann et al., 2023). However, this generalization capability is complex: PFNs might yield poor approximations even within the support, as they only approximate Bayesian predictions. Understanding when and why these approximation failures occur remains an open challenge.

Further of interest is analyzing how the sample-generating distribution's support of a specific dataset affects prediction quality. This relates to a broader phenomenon: while exact Bayesian inference converges to a nearby (in KL-div.) but necessarily wrong latent (Burt et al., 2020), neural networks have demonstrated an ability to gracefully model distributions that are similar to, but distinct from, their training distributions (Lake & Baroni, 2023; Raventos et al., 2023).

## 5.4. Interpretability of PFNs

PFNs present unique interpretability challenges compared to MCMC or VI, or also established domain-specific algorithms, such as random forests or linear regression for tabular ML, as they don't explicitly instantiate interpretable latents. We identify two key dimensions for improving PFN interpretability in future research.

**Dataset-Level Interpretability** focuses on explaining individual predictions by understanding how the prior and observed data interact. Key approaches include:

- Based on the knowledge available about the latent $\xi$ generating the dataset, train the PFN to predict the impact of features on predictions.
- Counterfactual analysis through systematic dataset modifications, e.g., computing Shapley values (Rundel et al., 2024; Muschalik et al., 2024).
- Allowing PFNs to return an approximation of the latent posterior it uses internally, as outlined in Section 4.4.
- A gradient-based assessment of the importance of individual *data points* on predictions. While this would be very costly or not possible for traditional methods, for PFNs a single backpropagation is enough to obtain data point importance indicators.

**Mechanistic Interpretability** examines how PFNs internally process priors and datasets not based on approximations, but by analyzing the model's state on particular inputs. Directions worth further exploration include:

- Analyzing internal representations of function classes and noise models, e.g., the question whether there are neurons representing the mean of latent variables or labels.
- Studying the attention distributions for datasets to understand how PFNs gather information.
- Developing architectures with interpretable components.
- Creating verification methods for properties like calibration.
- Finding commonalities and differences in the way PFNs handle data across different priors, seeds, architectures and training steps.

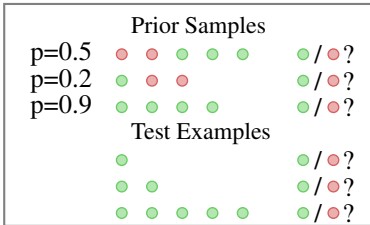 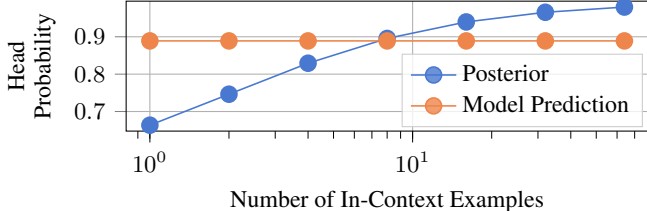

*Figure 3.* On the left (top), we outline our prior, sampling a probability $p$ for heads (green) and generating samples by coin flips. At test time (left, bottom), we condition on varying counts of coins displaying heads. On the right, we can see that the transformer-based PFN model is not able to approximate the posterior, as it would need to count the number of examples in the context, which are all identical.

# 6. Improving PFNs

The previous sections explored how we can extend and understand PFNs. In this section, we want to explore how we can improve current PFNs.

## 6.1. Scaling to More Samples

Despite the fact that current PFNs already handle up to 10,000 examples (Hollmann et al., 2025), scaling to larger datasets presents two main challenges.

**Compute** The quadratic scaling of transformer-based architectures with sample size can be mitigated with methods such as the Perceiver (Jaegle et al., 2021), state-space models (Gu & Dao, 2023), or linear attention (Wang et al., 2020a; Qin et al., 2022), which reduce the scaling to linear. While training models on more samples likely necessitates training on more datasets, TabPFN v2's pre-training efficiency (2,700 GPU hours) indicates a lot of potential for further computational scaling (Hollmann et al., 2025).

**Prior** Larger datasets generally diminish the influence of the prior in Bayesian models. This phenomenon occurs as the the posterior becomes more peaked with more evidence, due to the data likelihood dominating the influence of the prior as outlined for PFNs by Müller et al. (2024). Several factors suggest that this won't limit the effectiveness of PFNs soon, though. i) First, the success of ensembles in large-scale vision models (Kondratyuk et al., 2020; Wang et al., 2020b) hints at wider posteriors to yield benefits in large-scale settings. These ensembles approximate a posterior over weights that is not a dirac-delta, but wider. ii) Secondly, tree-based methods in large-scale tabular data (McElfresh et al., 2023) tend to outperform neural networks. Even though the neural networks are powerful and very likely able to fit the training set perfectly. This hints at the fact that the prior (inductive bias) underlying tree-based methods is better than that of neural networks, thus hinting that priors remain important when scaling the number of samples. For larger PFNs, we recommend using more general priors that support complex functions, enabling modeling of intricate functions with increased data.

Ensembling and fine-tuning PFNs post-pre-training offers further scalability (Thomas et al., 2024; Breejen et al., 2023; Feuer et al., 2024a), presenting a promising research avenue. Especially, fine-tuning might be a way to combine the generality and stability of PFNs with domain knowledge, in domains where a lot of related datasets are available, like forecasting or tabular prediction.

## 6.2. Focus Pre-Training on Hard Datasets

Presently, PFNs are trained directly by sampling from the prior. Most priors (Hollmann et al., 2023; Rasmussen & Williams, 2006) favor simple functions, aligning with Occam's razor principles. The probability a latent holds is proportional to the compute spent learning the datasets it represents, and thus common trained PFNs focus predominantly on simple datasets.

To address this, importance sampling could be employed. PFN training allows precise control over data generation, enabling hyperparameter adjustments in a well-defined parameter space, unlike when training on real-world data. A secondary model that accommodates distribution shifts, such as Drift-resilient TabPFN (Helli et al., 2024), could estimate gradient magnitudes across different prior settings, guiding a proposal distribution proportional to these magnitudes. Another proposal distribution source could be the model's performance improvements from training on specific prior regions; regions with greater improvements may yield further advancements. Additionally, given that language modeling faces similar issues, advancements in training efficiency through data direction could be applicable.

## 6.3. Fast Inference

PFNs face slow inference times due to their architecture combining dataset fitting with prediction. Three main acceleration approaches have emerged: caching training set states with multi-query attention (Hollmann et al., 2025; Shazeer, 2019), distilling datasets into fewer tokens (Feuer et al., 2024b), and using hypernetworks to predict an inference network (Müller et al., 2023b). Notably, caching can be viewed as predicting network weights. The caching in

Hollmann et al. (2025), for example, effectively functions as a hypernetwork predicting a transformer with attention across features. In detail, this predicted transformer has an additional sublayer per layer, though, which contains a set of two-layer MLPs using attention as activation function, whose outputs are concatenated and linearly projected. Future work should focus on architectural modifications for efficient caching and exploring offline approximation techniques like sparse attention through nearest neighbor search (Johnson et al., 2019).

### 6.4. Adaptive Compute Exertion

Not all predictions necessitate equal computational effort. Recent advancements in language models (Wei et al., 2022; DeepSeek-AI et al., 2025; Kimi et al., 2025) suggest PFNs can be enhanced for multi-step reasoning through: (i) iterative sampling with intermediate points (see Section 5.1) ideally situated between the query point and the training set, (ii) pre-training with variable-length causally masked tokens (Fan et al., 2019), incorporated as needed during inference, and (iii) reinforcement learning to optimize the computation-accuracy trade-off (DeepSeek-AI et al., 2025). This may also involve predicting a discretized "reasoning" path, offering insights into discrete "reasoning" processes.

### 6.5. Architectural Limitations

While PFNs aim to model Bayesian predictions (PPDs) for any prior, current architectures face two key limitations.

First, encoder-only transformers without positional embeddings struggle to count identical examples (Barbero et al., 2024; Yehudai et al., 2024). This limitation is clearly demonstrated in Figure 3, where the original PFN architecture (Müller et al., 2022) fails to process repeated inputs correctly. While TabPFN v2 (Hollmann et al., 2025) offers a workaround using inference-time noise features, more robust architectural solutions remain unexplored. A promising first step could be incorporating zero attention (`add_zero_attn` in PyTorch; Paszke et al., 2019).

Second, PFNs struggle with heterogeneous data distributions. Specifically, they perform poorly when mixing well-behaved centered distributions with heavy-tailed features, requiring prior knowledge of feature distributions for pre-processing. This limitation is particularly problematic for tabular prediction, where feature scales are often unknown beforehand. Future generations of (Tab)PFNs would greatly benefit from a novel encoder that automatically adapts to varying feature distributions.

## 7. Alternative Views

**View 1: Traditional Bayesian Methods are Superior** Critics may argue that traditional Bayesian meth-

ods, such as MCMC and VI, remain the gold standard for Bayesian inference due to their interpretability, access to the latent, and theoretical foundations. Furthermore, MCMC is correct in the compute limit.

**Response:** While MCMC and VI still dominate for Bayesian inference in general, we do not believe they will do so much longer for prediction tasks, for the reasons listed in Section 2.3.

**View 2: PFNs Do Not Scale** Critics may argue that PFNs are constrained by their quadratic scaling in sample size. This could prevent their application in domains requiring larger datasets, where methods like neural networks trained with stochastic gradient descent (or tree-based methods) may be more suitable.

**Response:** First, other Bayesian methods typically scale even worse (see Section 2.3), and multiple promising approaches exist to address PFNs' scaling limitations (see Section 6.1). We do believe, though, that PFNs will not be the solution to all machine learning problems. Large-scale problems, such as language modeling, will likely always rather profit from a non-Bayesian treatment.

**View 3: PFNs Lack Interpretability** Critics may argue that PFNs sacrifice the interpretability of traditional Bayesian methods by not explicitly modeling the latent.

**Response:** While this poses a challenge for PFNs, Section 4.4 outlines possibilities for modeling the latent space, and Section 5.4 outlines numerous promising approaches to enhance PFN interpretability, including dataset-level and mechanistic interpretability methods. Moreover, as discussed in Section 3, PFNs' declarative nature allows practitioners to explicitly encode domain knowledge through prior specification, providing a different but valuable form of interpretability.

## 8. Conclusion

In this position paper, we argue that PFNs represent a transformative approach to Bayesian prediction. By enabling declarative programming through prior specification and synthetic pre-training, PFNs make sophisticated Bayesian methods both accessible and computationally efficient. Their effectiveness across domains, from tabular to genetics data, and particular strength in data-scarce scenarios position them as a robust alternative to traditional Bayesian methods and a promising avenue towards probabilistic foundation models. While challenges remain, promising research directions like in-context interpreters and latent prediction methods suggest that PFNs will become increasingly central to probabilistic machine learning.

## Acknowledgements

We are grateful for the computational resources that were available for this research. Specifically, we acknowledge support by the state of Baden-Württemberg through bwHPC and the German Research Foundation (DFG) through grant no INST 39/963-1 FUGG (bwForCluster NEMO), and by the Deutsche Forschungsgemeinschaft (DFG, German Research Foundation) under grant number 417962828. Frank Hutter acknowledges the financial support of the Hector Foundation.

We acknowledge funding through the European Research Council (ERC) Consolidator Grant "Deep Learning 2.0" (grant no. 101045765). Funded by the European Union. Views and opinions expressed are however those of the author(s) only and do not necessarily reflect those of the European Union or the ERC. Neither the European Union nor the ERC can be held responsible for them.

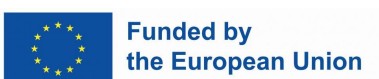

## Impact Statement

This paper presents work whose goal is to advance the field of Machine Learning. There are many potential societal consequences of our work, but we would like to highlight a few. First, we believe that PFNs might (i) reduce the $CO_2$ cost for a particular application of Bayesian prediction, but (ii) it might also increase the number of applications that use Bayesian prediction, rather than methods with lower $CO_2$ cost.

Further, PFNs equip researchers and data scientists with more proper, well-calibrated uncertainty estimates, which should lead to better decisions downstream. These decisions are probably the main contributor to positive and negative effects in the world. We believe that better estimates lead to better outcomes in the world for such general methods.

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

# A. Experimental Setup

We conducted all experiments using the original PFN architecture (Müller et al., 2022). A grid search identified the model with the best final training loss. We searched across 4 and 8 layers, batch sizes of 32 and 64, Adam learning rates of 0.0001, 0.0003, and 0.001, embedding sizes of 128, 256, and 512, and step counts of 100 000, 200 000, and 400 000. Training set sizes were uniformly sampled from 1 to 100.

## A.1. Priors

**GP Prior for Section 5**    For the analysis on the Martingale property in Section 5, we utilized the standard GP proposed by Müller et al. (2022) with an RBF-Kernel, length scale of 0.1, output scale of 1.0, and noise standard deviation of $10^{-4}$. Inputs were uniformly sampled between 0 and 1.

**Coin Flipping Prior for Section 6.5**    To illustrate the PFN architecture's limitation in counting duplicated samples, we trained a PFN on random coin flips in Section 6.5. This simple prior asks the model to predict a coin's probability of landing heads, where a coin with a different head probability uniformly chosen from $\{0.01, 0.02, \ldots, 0.99\}$ is sampled to generate each dataset. As shown in Figure 3, when feeding only samples that landed heads, the neural network's prediction remains static despite the posterior's expected evolution to assume a higher probability of head. This demonstrates the network's failure to update its beliefs with new evidence, as it can't count duplicated samples.

# B. Martingale Properties of PFNs

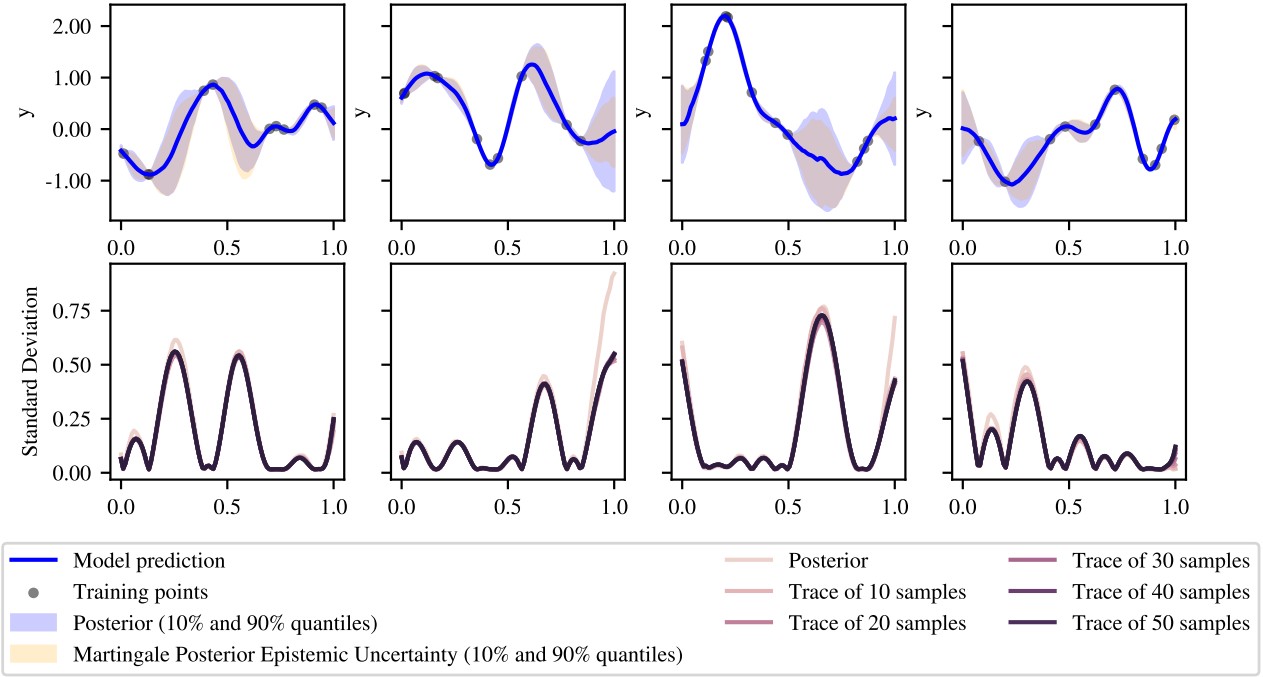

*Figure 4.* We train a model on two distinct classes of functions, sines and sloped lines, only (left). It not only learns fit both function classes well (center), but also learns to model slightly sloped sines, when prompted with a data from a sloped sine.

