# OpenReview forum: "Position: The Future of Bayesian Prediction Is Prior-Fitted"
_ICML.cc/2025/Position_Paper_Track — ICML 2025 Position Paper Track poster_

### Official Review · Reviewer_Z1sn · 2025-03-03

**Significance:** 2
**Argument Clarity:** 3
**Rating:** 3
**Confidence:** 4

**Questions:**

See Strengths And Weaknesses.

**Discussion Potential:**

3

**Paper Summary:**

The paper advocates for the use of prior data-fitted networks (PFNs) for approximate Bayesian predictions. The authors first present background on PFNs and relationships to other Bayesian modeling options. Then, the authors discusses several aspects of PFNs such as its current limitations and possible remedies for them, possible extensions for this family of model classes, and several directions for understanding their behavior.

**Position:**

Yes

**Position In Title:**

Yes

**Related Work:**

2

**Strengths And Weaknesses:**

Strengths:
- The paper introduces an interesting research direction that can be impactful in certain learning scenarios as demonstrated in previous studies.
- The topic of the paper is important for the ICML and Bayesian communities. The proposed approach tries to address known limitations of current Bayesian deep learning approaches.
- The paper is written well and easy to follow.

Weaknesses/Questions:
- Can the authors please elaborate more on the importance of the quality of the synthetic data and the training method? Specifically, how close it should be to the real data of interest. To me, it seems that if the distribution of the synthetic datasets differs too much from the real one the pre-training stage will not help much (as evident in the domain adaptation literature). On the other hand if one can generate synthetic data that closely matches the real data of interest, then why do the proposed type of training and not standard NN training using all the simulated data (as one big dataset)?
- The link to the Bayesian paradigm is also not clear to me. A key property of the Bayesian approach is to apply marginalization over some quantity when making predictions (and hence capture epistemic uncertainty) [1]. Here predictions are made by doing a forward pass while conditioning on the training set, but this does not constitute as doing a Bayesian model averaging as far as I can tell. Can the authors please clarify that point?
- Regarding the training objective, how do the results change if instead of optimizing for the predictive likelihood one does standard NLL training (using a similar architecture and training procedures)? In some Bayesian meta-learning methods (e.g., [2, 3]) the two approaches were evaluated and there was no clear advantage to either one. Does the proposed approach work better?
- I believe that the method proposed in [4] is relevant to this study as a related approach. Although not working with synthetic data, in [4] the authors discuss a similar objective to that proposed in this study and show that it tends to overfit. They also proposed a solution by matching the predictive posterior of the model to that of a "pure" Bayesian model, using it as prior. Could this strategy be used in the case discussed in this paper as well?
- How do the uncertainty quantification abilities of PFN on real-world data? The authors didn't address that point which is one of the main goals in Bayesian modeling. My intuition (and I may be wrong) is that these models tend to be underconfident as they do not rely on the training set of the real data for adjusting the model.


[1] Wilson, A. G., & Izmailov, P. (2020). Bayesian deep learning and a probabilistic perspective of generalization. Advances in neural information processing systems, 33, 4697-4708.
[2] Snell, J., & Zemel, R. (2020) Bayesian Few-Shot Classification with One-vs-Each Pólya-Gamma Augmented Gaussian Processes. In International Conference on Learning Representations.
[3] Achituve, I., Shamsian, A., Navon, A., Chechik, G., & Fetaya, E. (2021). Personalized federated learning with Gaussian processes. Advances in Neural Information Processing Systems, 34, 8392-8406.
[4] Achituve, I., Chechik, G., & Fetaya, E. (2023, July). Guided deep kernel learning. In Uncertainty in Artificial Intelligence (pp. 11-21). PMLR.

**Support:**

2

---

> ### Author Rebuttal · Authors · 2025-03-30
>
> Thank you for your review. We can address your concerns and will update our paper to clarify all points in this rebuttal.
>
> # Strengths
>
> You write
>
> > The proposed approach tries to address known limitations of current Bayesian deep learning approaches.
>
> To clarify: PFNs are not Bayesian neural networks with uncertainties over their weights. The PFN itself is a classical neural network that accepts a training set and a test input, and returns approximations of the posterior predictive distribution for the prior from which its pre-training datasets are drawn.
>
> # Weaknesses/Questions
>
> ## Point 1
>
> You write
>
> > Can the authors please elaborate more on the importance of the quality of the synthetic data and the training method?
>
> It is important that similar datasets are covered in the prior.
> As shown in this [analysis paper], it is enough that a similar dataset has some probability to be sampled during pre-training, where higher likelihoods being better.
>
> You further write
>
> > why ... not standard NN training using all the simulated data (as one big dataset)?
>
> We do not follow this approach, as it tends to be simpler to define broad priors that cover many datasets. Traditional GP priors are a good example for this: The datasets sampled from them, see Figure 3 of this [GP lecture], are all kinds of smooth datasets. If one trains a PFN on this, it approximates the GP predictions which is good for smooth datasets. Try it in this [demo].
>
> ## Point 2
>
> You write
>
> > The link to the Bayesian paradigm is also not clear to me. A key property of the Bayesian approach is to apply marginalization...
>
> PFNs are indeed a Bayesian approach, but they only implicitly marginalize over latents, as they directly approximate the PPD $p(y|x,D) = \int p(y|x,\xi) p(\xi|D) d\xi$, which itself marginalizes over the latent $\xi$.
> The (pre-)training loss of PFNs can be written as
>
> $$L(\\theta) = E_{\\xi \sim p(\\xi); (x,y) \sim p(x,y|\\xi); D \sim p(D|\\xi)}[-\log(q_\\theta(y|x,D))],$$
>
> where $D$ is a set of $(x,y)$ pairs sampled as $(x,y) \sim p(x,y|\xi)$, and $q_\theta$ is the PFN model. Cor. 1.1 of [PFNs] shows this is equal to
>
> $$L(\\theta) = E_{(x,D)\sim p(x,D)}[KL(p(y|x,D)||q_\theta(y|x,D))]+C$$
>
> where KL denotes the KL divergence and C denotes a constant, independent of $\\theta$.
> Thus, PFNs are trained to minimize the KL divergence between the true (marginalized) PPD and the prediction across data sampled from the prior.
>
> You further write:
>
> > ...this does not constitute as doing a Bayesian model averaging...
>
> In a recent [analysis paper], the authors elucidate the point that PFNs actually do implicitly perform Bayesian model averaging experimentally. In Section 4 of the [analysis paper] the model is able to generalize to out of distribution data via implicit Bayesian model averaging.
>
> ## Point 3
>
> You write
>
> > how do the results change if instead of optimizing for the predictive likelihood one does standard NLL training
>
> PFNs are actually trained with standard NLL. However, they predict the outcome not only based on the current sample’s features but additionally condition on the whole dataset. The training is performed across a large number of synthetic, sampled datasets.
>
> Could you please clarify this question.
>
> You further recommend the approaches from [2] and [3].
> Both papers focus on few-shot / meta-level Bayesian models. They are Bayesian on the meta-level and target tasks with multiple datasets, but PFNs perform classical Bayesian supervised learning.
>
> ## Point 4
>
> You write
>
> > ... [4] is relevant to this study as a related approach ... [they] discuss a similar objective to that proposed in this study and show that it tends to overfit.
>
> [4] proposes a method to regularize the training of Deep Kernel GPs. Unlike them, PFNs are trained on an infinite set of (synthetic) datasets.
> Thus, PFNs do not tend to overfit, which is exemplified in outstanding calibration of [TabPFNv2, Extended Data Table 1] and strong BayesOpt performance [PFNs4BO,LC-PFN,FT-PFN].
>
> ## Point 5
>
> You write
>
> > How do the uncertainty quantification abilities of PFN on real-world data?
>
> There is evidence that uncertainties are good for real-world data, if the right prior is chosen. See the point above for references.
> PFNs can,e.g., be underconfident if the prior has a high entropy, similar to GPs being underconfident, if one assumes large label noise.
>
> [GP lecture]
> https://www.cs.cmu.edu/~16831-f12/notes/F12/16831_lecture20_venkatrn.pdf
>
> [demo]
> https://huggingface.co/spaces/samuelinferences/transformers-can-do-bayesian-inference
>
> [PFNs]
> https://openreview.net/pdf?id=KSugKcbNf9
>
> [analysis paper]
> https://arxiv.org/abs/2410.01565
>
> [TabPFNv2]
> https://www.nature.com/articles/s41586-024-08328-6
>
> [PFNs4BO] https://proceedings.mlr.press/v202/muller23a.html
>
> [LC-PFN]
> https://proceedings.neurips.cc/paper_files/paper/2023/hash/3f1a5e8bfcc3005724d246abe454c1e5-Abstract-Conference.html
>
> [FT-PFN]
> https://dl.acm.org/doi/abs/10.5555/3692070.3693777

---

> > ### Comment · Reviewer_Z1sn · 2025-04-04
> >
> > I am sorry for the late reply, I wrote a comment a couple of days ago but it wasn't visible to you. In any case, thank you for the answers. Most of the points made by me and other reviewers were properly addressed; hence, I raised my score to 3.

---

> > > ### Author Response · Authors · 2025-04-04
> > >
> > > We are happy we could clarify things in our rebuttal and will update the camera-ready version accordingly.
> > >
> > > Please, let us know if you have more questions or comments that are open.

---

### Official Review · Reviewer_2Ffb · 2025-03-13

**Significance:** 3
**Argument Clarity:** 3
**Rating:** 3
**Confidence:** 3

**Questions:**

Can you clarify what the potentially fundamental limitations in learning from large-scale data in context are (page 4)?

The preliminary evaluationt that PFNs fail to satisfy the Marginale property is interesting. It would be interesting what this implies for the stated positions that PFNs are the future for Bayesian inference. In particular, while traditional Bayesian inference methods (assuming a good variational approximation, long enough MCMC chains) should satisfy the marginale property, do you have a suggestion why PFNs do not satisfy it, such as due to the pre-training nature, or more because of the model architecture limitation.

The authors argue that PFNs struggle with heterogeneous data. Can this be accounted for by including synthetic data with such distributional features into the pre-training dataset? As written, it sounds like this is more due to the encoder not able to adapt to the varying feature distribubutions due to architectural limits.

**Discussion Potential:**

3

**Paper Summary:**

The authors presents the position that synthetic-data pre-training, with PFNs being the classic example for tabular data, represents the future of Bayesian prediction. The paper contrasts 'standard' Bayesian approaches that rely on commonly explicit priors versus prior-fitted ones that are trained on synthetic data. Similarities and difference to related Bayesian approaches such as SBI or Neural Processes are discussed. The paper summarises PFN extensions (e.g. incorporating extra inputs) and limitations (e.g.limited uncertainty with a factorised distribution over the outputs, scalability for large datasets, slow inference time, difficult to interpret in absence of performing inference on latent concepts). The paper also suggests some approaches to address limitations of PFNs.

## update after rebuttal
The authors have addressed most of my concerns and questions in the rebuttal.

**Position:**

Yes

**Position In Title:**

Yes

**Related Work:**

3

**Strengths And Weaknesses:**

The authors argue that PFNs 'dramatically expand the space of possible priors that can be practically used'. It would be useful to clarify how this relates to other generative models, that commonly rely on an Empirical Bayes approach to learn prior distributions, that I would argue are also quite flexible, e.g. via diffusion models. Likewise, the authors argue that PFNs 'only require the ability to sample from the prior'. However, implicit generative models [1] would also satisfy these desiderata, but are usually estimated via empirical Bayes. I feel that the concept of pre-training should be presented more closely as a difference to such models.

The paper argues that PFNs allow for the direct encoding of assumptions by including corresponding examples during pre-training. But does this then require learning a new parameter $\theta$ for each such case, somewhat defeating the generalisability of the method. Essentially, I feel that the paper would benefit from some discussion about fine-tuning approaches in such a setup that may address these issues and how it relates to (and possibly improves?) Meta-learning approaches with a similar Bayesian framework such as [2,3], or more recent Bayesian fine-tuning approaches for foundation models such as  [4,5].

The paper introduces (?), including some pseudo-code in the main paper, PFNs as in-context interpreters. There appear no empirical evaluation for this and I am not sure if this fits the position paper format.

The outlines potential approaches how to adress limitations of PFNs are interesting.

The paper is of relevance to the community. Alternative views are discussed.


[1] Mohamed, Shakir, and Balaji Lakshminarayanan. "Learning in implicit generative models." arXiv preprint arXiv:1610.03483 (2016).
[2] Grant, E., Finn, C., Levine, S., Darrell, T., & Griffiths, T. (2018, February). Recasting Gradient-Based Meta-Learning as Hierarchical Bayes. In International Conference on Learning Representations.
[3] Ravi, S., & Beatson, A. (2019, May). Amortized bayesian meta-learning. In International Conference on Learning Representations.
[4] Pandey, Deep, Spandan Pyakurel, and Qi Yu. "Be Confident in What You Know: Bayesian Parameter Efficient Fine-Tuning of Vision Foundation Models." Advances in Neural Information Processing Systems 37 (2024): 44814-44844.
[5] Kim, Minyoung, and Timothy Hospedales. "LiFT: Learning to Fine-Tune via Bayesian Parameter Efficient Meta Fine-Tuning." The Thirteenth International Conference on Learning Representations.

**Support:**

3

---

> ### Author Rebuttal · Authors · 2025-03-31
>
> Thank you for your review and positive feedback.
>
> ## Strengths and Weaknesses
>
> ### Point 1
>
> You write
> >  It would be useful to clarify how this relates to other generative models that commonly rely on an Empirical Bayes approach to learn prior distributions [...] e.g. via diffusion models
>
> PFNs are not an Empirical Bayes approach—the prior is pre-defined, and they are discriminative. They provide an approximation to classical Bayesian posterior predictives, like MCMC or VI, in a supervised setting.
> We mention Diffusion Models in Sec. 2.2 and add more analysis of how to use them to model, e.g., the latents (Sec. 4.4).
>
> ### Point 2
>
> You write
>
> > the authors argue that PFNs 'only require the ability to sample from the prior'. However, implicit generative models [1] would also satisfy these desiderata [...] I feel that the concept of pre-training should be presented more closely as a difference to such models.
>
> The difference to implicit generative models is that PFNs are not generative, but discriminative with a hand-defined prior. The prior might, e.g., be to draw smooth curves from a GP prior. (Refer to Figure 1).
> We will add more details on the pre-training.
>
> ### Point 3
>
> You write
> > The paper argues that PFNs allow for the direct encoding of assumptions by including corresponding examples during pre-training. But does this then require learning a new paramete for each such case
>
> Could you clarify this, please? The question might stem from misunderstanding PFNs as generative models.
> PFNs are trained on hand-designed dataset generators, and one might, e.g., add a lot label noise on 1% of datasets such that the model can handle noisy scenarios.
>
> ### Point 4
>
> You write
> > the paper would benefit from some discussion about fine-tuning approaches
>
> We already mention three works on fine-tuning TabPFN (end of Sec. 6.1), and will add more discussion of fine-tuning there.
>
> You further write
>
> >  Meta-learning approaches with a similar Bayesian framework such as [2,3], or more recent Bayesian fine-tuning approaches for foundation models such as [4,5].
>
> The approaches in [2,3,5] are concerned with a Bayesian approach to meta-learning, while we are only performing classical Bayesian supervised learning. The pre-training on many datasets can be seen as part of method development, we do not have a meta-prior across datasets.
> [4] is a work to enable more calibrated fine-tuning that could definitely be useful for PFN fine-tuning such as in [TuneTables,LoCalPFN].
>
> ### Point 5
>
> You write
>
> > The paper introduces […] PFNs as in-context interpreters [...] I am not sure if this fits the position paper format.
>
> We are happy to remove this, if you prefer. The idea for this is to draw a clearer picture of the future that PFNs enable in order to strengthen our argument that they will become dominant for Bayesian predictions.
>
> ## Questions
>
> ### Point 6
>
> You write
>
> > Can you clarify what the potentially fundamental limitations in learning from large-scale data in context are (page 4)?
>
> Sure. Section 6.1 already has more details than page 4 and we will extend this section further.
> The potentially fundamental limitation is that PFNs likely benefit from being Bayesian for smaller datasets more than for larger datasets, as the posterior becomes more peaked with more evidence. Thus, a maximum a-posteriori point estimate might perform well enough for large datasets.
> There is, however, some counter evidence:
> i) ensembles of NNs perform well even for large datasets [Ens1,Ens2], and
> ii) tree-based methods are stronger than NNs for many large datasets [Trees-vs-NNs], which likely is due to their inductive bias, translating to the prior in Bayesian settings.
>
> ### Point 7
>
> You write
>
> >  It would be interesting what [the failing martingale property] implies for the stated positions
>
> The martingale property is broken only sometimes, especially on the ends of the interval (see Appendix B). We believe this is due to modelling inaccuracies, which should vanish with improving models.
> There are no guarantees with the current PFN architectures, though, and they might still break for o.o.d. data. However, there might be ways to fix the Martingale property architecturally.
>
> ### Point 8
>
> You write
>
> > The authors argue that PFNs struggle with heterogeneous data. Can this be accounted for by including synthetic data with such distributional features into the pre-training dataset?
>
> Yes, this is one way to counter this and done in [TabPFNv2]. The encoder is important, too, though: it might, e.g., be hard for a linear encoder [TabPFNv2] to train if .001% of inputs are 1 million times larger.
>
> [TabPFNv2] https://www.nature.com/articles/s41586-024-08328-6
>
> [Ens1] https://arxiv.org/abs/2005.00570
>
> [Ens2] https://arxiv.org/abs/2012.01988
>
> [Trees-vs-NNs] https://proceedings.neurips.cc/paper_files/paper/2023/hash/f06d5ebd4ff40b40dd97e30cee632123-Abstract-Datasets_and_Benchmarks.html

---

> > ### Comment · Reviewer_2Ffb · 2025-04-02
> >
> > I thank the authors for their rebuttal that clarified some points.
> >
> > >The paper argues that PFNs allow for the direct encoding of assumptions by including corresponding examples during pre-training. But does this then require learning a new paramete for each such case
> > >>Could you clarify this, please? The question might stem from misunderstanding PFNs as generative models. PFNs are trained on hand-designed dataset generators, and one might, e.g., add a lot label noise on 1% of datasets such that the model can handle noisy scenarios.
> >
> > To clarify this point: This is not related to a generative capacity, but that it is not clear to me how to leverage the in-context learning capacity on (publicly available pre-trained) PFNs when one has specific assumptions on the data (e.g. the mentioned Gamma-distributed noise). Ideally, I would not like to learn new parameters of the PFN by pre-training 'again' whenever I add these corresponding examples to the pre-training dataset. Instead, I may like to perform some meta-learning or fine-tuning on a pre-trained model, and from this perspective, I was wondering how PFNs can be beneficial or not compared to the referred  Bayesian Meta-learning or fine-tuning methods [2-5].

---

> > > ### Author Response · Authors · 2025-04-03
> > >
> > > That is a very good point. This indeed is a downside in the way PFNs are trained so far: You have to redo pre-training whenever you change the prior.
> > >
> > > We believe there are two ways to lessen this problem effectively:
> > >
> > > i) Based on what you write: fine-tuning on the new prior like you describe to give the model a good starting point to improve. We will explicitly add that to the paper. Do we understand your idea right, that you propose training PFNs to be good at fine-tuning in a MAML-style? That does indeed sound very interesting and we would like to add this idea, if we may.
> > >
> > > ii) Training PFNs across multiple priors at once, as described in Section 4.1, and brought to its extreme of freely programmable PFNs in Section 4.2.

---

### Official Review · Reviewer_tNBT · 2025-03-14

**Significance:** 3
**Argument Clarity:** 2
**Rating:** 4
**Confidence:** 3

**Questions:**

In equation 1, what is $\phi$? Is it a typo?

**Discussion Potential:**

3

**Paper Summary:**

This paper argues that prior-fitted networks deserve more attention and will replace other Bayesian prediction methods in the future. The argument is essentially that they can handle complex distributions, can use "unlimited" training data, and don't require expensive sampling (NOTE: other reasons were mentioned too).

### update after rebuttal

The authors answered some of my concerns about the scope of the paper (PFNs vs all of generative AI). I changed my score 3 -> 4.

**Position:**

Yes

**Position In Title:**

Yes

**Related Work:**

3

**Strengths And Weaknesses:**

**Strengths**: the paper provides a good overview of PFNs, honestly discusses their limitations, and provides a good summary of their strengths/weaknesses relative to traditional Bayesian methods (with some slight issues though, see point 5 below). It addresses alternative views and could inspire discussion at ICML. The discussion of related work was very fair and reasonably comprehensive (although with some potentially large omissions, see point 1 below). The paper was very well-written; overall great job!

**Weaknesses**: although I liked the paper overall, I had a few concerns, mostly about the scope of the position taken:

1. Are PFNs the same as meta-learning (or even "generative AI" more broadly)? The discussion on neural processes makes me think that the answer is _yes_. The authors differentiate PFNs from NPs by stating that NPs are usually trained on a fixed dataset and that PFNs are trained on randomly sampled datasets, but in NPs/meta-learning people often _do_ train on randomly sampled datasets. Furthermore, if you re-label $(x,y):=z$, then PFNs are essentially the same as the kind of masked maximum likelihood training from LLMs. It feels like the position should be expanded to cover these methods too, no? In fact, I almost think the logical extension of the argument is  "let's just do generative AI instead of defining Bayesian models"
2. This feels more like a review paper than a position paper. The position is _almost_ "the field of PFNs has grown a lot in the past few years". Most of the paper is about explaining + reviewing PFNs rather than arguing for the position (although I acknowledge that part of the argument is the reviewing)
3. Specifying the data-generating distribution in PFNs seems like a very different way of being "Bayesian" than traditional methods (eg Bayesian parametric models). With traditional methods, one specifies a data-generating mechanism via a model, then performs inference over the model's latents. In PFNs it seems like datasets are randomly sampled, possibly in a model-free way? Maybe it would help to explain this more in the paper. In any case, it seems a bit strange to say that PFNs will replace other Bayesian methods when they are Bayesian in very different ways.
4. PFNs seem limited by the ability of the model to capture the actual posterior predictive distribution. For example, if the true posterior predictive is continuous and bimodal it seems unlikely to me that *any* transformer model will be able to capture this (although I might be wrong here). In contrast, traditional (exact) Bayesian methods are guaranteed to be able to model the posterior predictive because it is defined in terms of the model itself.
5. I found the interpretation of traditional Bayesian methods a bit odd. It contrasts PFNs to MCMC and VI, but these are only _inference_ methods, they need to be paired with a model. The more appropriate comparison is the model + approximate inference method, and different approximate inference methods will work better or worse depending on the model. In that sense, it seemed odd to describe MCMC/VI in isolation.

Overall I think this paper probably meets the bar for acceptance, but I think it would be better if the scope was changed slightly to include highly-related methods, and if the philosophical differences to traditional Bayesian methods were explained as part of the argument.

**Support:**

3

---

> ### Author Rebuttal · Authors · 2025-03-31
>
> Thank you for your thorough review and positive feedback.
>
> ## Point 1
>
> You ask
>
> > Are PFNs the same as meta-learning (or even "generative AI" more broadly)?
>
> The PFN methodology builds upon meta-learning. However, we think that the central component of PFNs is training on synth. datasets sampled from a prior, which makes PFNs an approximation to the PPD of that prior, see, e.g., this [demo]. We argue that neither LLMs nor generative AI at large nor NPs share this property, and it is neither a property present in other meta-learning approaches.
> PFNs are not a specific type of architecture, but rather a specific training paradigm.
>
> Later you come back to the scope
>
> > I think it would be better if the scope was changed slightly to include highly-related methods, and if the philosophical differences to traditional Bayesian methods were explained as part of the argument.
>
> We think arguing for generative AI in general is beyond the scope of this paper and we will make the distinction (see above) to these methods clearer.
> We further think that a more philosophical perspective on the differences to traditional Bayesian inference are interesting and we will further detail the relationship of prior-fitting and model-based approaches.
>
> ## Point 2
>
> You write
> > This feels more like a review paper than a position paper.
>
> We think that explaining PFNs and reviewing recent work is crucial for motivating our position to a broad readership. We do not agree that this is most of the paper, though. On pages 3 and 4, we review the relevant literature, but only alongside providing reasons for PFNs to become dominant, and we commit most of the second half of the paper (pages 5 to 8) to discuss future topics to show how PFNs can become dominant.
>
> ## Point 3
>
> Your write
> > In PFNs it seems like datasets are randomly sampled, possibly in a model-free way? Maybe it would help to explain this more
>
> That is right. There are many usages of PFNs with GP priors [PFNs,PFNs4BO], which do not instantiate a model.
> We are currently working on making this clearer, similar to [PFNs], and will further outline the relationship to marginalization over latents which PFNs implicitly approximate; see Point 2 in Z1sn's rebuttal.
>
> You further write
> > In any case, it seems a bit strange to say that PFNs will replace other Bayesian methods when they are Bayesian in very different ways.
>
> They do work in a very different way, but they still maintain the same interface and approximate the same thing (the PPD). Just like MCMC works very differently from Variational Inference. Our position is that precisely their special and different characteristics make them promising. (See also Point 5)
>
> ## Point 4
>
> You write
> > PFNs seem limited by the ability of the model to capture the actual posterior predictive distribution. [..] if the true posterior predictive is continuous and bimodal it seems unlikely to me that any transformer model will be able to capture this
>
> PFNs are limited by the modeling abilities of the neural net, but neural nets are powerful. Exactly a multimodal distribution is predicted by TabPFN in Figure 3b of [TabPFNv2]. PFNs typically use a discretized continuous distribution, so they can model multi-modality.
> In the paper proposing PFNs [PFNs], it was further shown that in the limit of an expressive enough model and training to the global optimum in pre-training, they do indeed exactly model the PPD.
>
> You further write
>
> >  In contrast, traditional (exact) Bayesian methods are guaranteed to be able to model the posterior predictive because it is defined in terms of the model itself.
>
> We would like to argue that the set of practically relevant and exact Bayesian methods is very limited beyond simple Gaussian processes with a fixed kernel and linear models. In more complex models, Bayesian predictions are typically performed via an approximation such as MCMC or VI. Analogous to PFNs, MCMC and VI are not exact.
>
> ## Point 5
>
> You write
> > I found the interpretation of traditional Bayesian methods a bit odd. It contrasts PFNs to MCMC and VI, but these are only inference methods
>
> With PFNs we mean the general methodology of training neural nets to predict hold-out labels on datasets sampled from a prior. They yield an approximation of the PPD on new datasets making them an inference method—or rather a prediction method.
> One can for example use a Bayesian neural network prior (sample random NN weights for each dataset). The archictecture of the BNN will be independent of the PFNs' architecture.
>
> ## Question
>
> You write
> > In equation 1, what is? Is it a typo?
>
> Yes, this is indeed a typo. Thank you! The corrected version is:
> $$E_{(\xi, x)\sim p(\xi, x)}[-\log q_{\theta}(\xi|x)].$$
>
>
> [demo] https://huggingface.co/spaces/samuelinferences/transformers-can-do-bayesian-inference
>
> [PFNs] https://openreview.net/pdf?id=KSugKcbNf9
>
> [PFNs4BO] https://proceedings.mlr.press/v202/muller23a.html
>
> [TabPFNv2] https://www.nature.com/articles/s41586-024-08328-6

---

> > ### Comment · Reviewer_tNBT · 2025-04-03
> >
> > I read your responses and the responses to the other reviewers. Overall:
> >
> > - I see "synthetic data" as core to the PFN framework and what differentiates it from traditional meta-learning.
> > - I acknowledge that this papers argues for a position (and is not just a "review" paper)
> > - I see how PFNs can be viewed as "approximate inference techniques" rather than just "a different type of model"
> >
> > I think the main thing I am missing is the details of the synthetic data generation. I get that in some scenarios you might have a black-box sampler (eg weather prediction simulations). _How do PFNs usually generate synthetic data, for example for tabular data?_ You acknowledged [here (point 1)](https://openreview.net/forum?id=5Hpm74b1Ga&noteId=RDC7BZYGKw) that similarity between synthetic and real data is critical.
> >
> > For what it's worth, if the "synthetic data" is just subsampling real datasets then the data doesn't seem "synthetic", and the approach seems basically the same as meta-learning.
> >
> > In any case, my opinion of the paper certainly has not gone _down_, so I am happy to keep my score. Additionally, I will consider raising my score after your response to my question above.

---

> > > ### Author Response · Authors · 2025-04-03
> > >
> > > Thank you for thoroughly reading our rebuttal(s).
> > >
> > > That is right: The synthetic data generation is the crucial point of PFNs. Besides cases where there is a black-box sampler available, the PFN literature typically uses broad priors, similar to the priors used for other Bayesian methods for the corresponding applications.
> > >
> > > ### Examples of Priors
> > > We list three examples that we think might resonate with you.
> > > We will add examples to the paper as well, to make it more concrete what priors look like for PFNs.
> > >
> > > **A GP prior for Bayesian Optimization**
> > > In the [PFNs4BO] paper, the task is to perform Bayesian optimization (BO) with PFNs. Gaussian Processes (GP) are very popular for BO, thus this paper uses a GP prior, based on the [HEBO] method. This prior is a GP prior with "meta-prior" over GP hyperparameters (length scale etc.).
> > >
> > > This sampling of one dataset in the pre-training is done as follows (+ some extra pre-processings you can find in the paper [PFNs4BO]).
> > > 1. Sample the hyper-parameters $\gamma$ of the GP from the hyper-prior, e.g. a uniform distribution over length scales in some interval.
> > > 2. Sample x's i.i.d. uniformly at random in [0,1]^d: $X \sim U([0,1]^d)$, where $X$ is a matrix with a row for each sample in the dataset and $d$ columns.
> > > 3. Sample from the GP, with the sampled hyper-parameters at the x positions to generate the targets y: $y \sim \mathcal{N}(\mathbf{0}, K)$, where $K$ is the kernel matrix, build based on the kernel function $K_{i,j} = k_\gamma(X_i,X_j)$
> > >
> > > You can see that this process is very general and will sample diverse datasets, see Figure 3 of this [GP lecture handout] for some examples of what 1d GP samples with different kernels look like.
> > >
> > > **A Bayesian Neural Network prior for Bayesian Optimization**
> > > The [PFNs4BO] paper further has a Bayesian Neural Network (BNN) prior, for a BNN with multi-layer perceptron (MLP) architecture. BNNs are neural networks with uncertainty over their weights and their prior is typically defined in terms of the prior distribution over their weights $\phi$ of the NN. In this paper it is assumed that they are drawn i.i.d. as $\phi \sim \mathcal{N}(\mathbf{0},\sigma)$.
> > > Similar to the previous prior, the (uninformative) uniform prior over x is used.
> > >
> > > So, the main part (besides some pre-processing) of the generation of one pre-training dataset looks like this:
> > > 1. Sample the MLP parameters i.i.d. $\phi \sim \mathcal{N}(\mathbf{0},\sigma)$.
> > > 2. Sample x's i.i.d. uniformly at random in [0,1]^d: $X \sim U([0,1]^d)$, where $X$ is a matrix with a row for each sample in the dataset and $d$ columns, like before.
> > > 3. Feed each row of $X$ through the neural network with weights $\phi$ to get the corresponding label $y$ as output of the neural network.
> > >
> > > Similar to above, this prior is very wide, as it covers all functions that the chosen architecture can represent. Next, it will be made a little more focused on a relevant subset.
> > >
> > > **The TabPFN Prior**
> > > The [TabPFNv2] most likely has the most sophisticated prior of all PFNs so far. It is an extension of the MLP prior, where extra noises are added to intermediate neurons, as well as a variety of activation functions being used in a single network. Further, it employs discretizations of nodes in some places, to allow for representing datasets with categorical features, and it uses a lot of sparsity in the layers.
> > > It further does not use the output of the network as target and the inputs as features, but instead samples random positions in the network for these instead. That is why they do not refer to it as an MLP/BNN prior, but rather as a stochastic causal model (SCM) prior, which is a generalization of MLPs used in the causal literature.
> > >
> > > The TabPFN prior is likely as general as the BNN prior, but with a lot of work put into increasing the likelihood of relevant scenarios, e.g. categorical features that only take on integer values.
> > >
> > >
> > > **Domain Priors** There are also works that build priors that are more targeted on particular domains, e.g., the prior in [LC-PFN] which is designed to mimic learning curves seen in machine learning workloads or the prior in [ForecastPFN], which is a prior over time-series for forecasting and includes seasonal behaviors that align with the calendar, e.g. weekly periods are likely.
> > >
> > >
> > > We hope this clarifies your questions regarding prior design and we are happy to answer any more questions or follow-ups you might have.
> > >
> > >
> > > [PFNs4BO] https://proceedings.mlr.press/v202/muller23a.html
> > >
> > > [GP lecture handout] https://www.cs.cmu.edu/~16831-f12/notes/F12/16831_lecture20_venkatrn.pdf
> > >
> > > [TabPFNv2] https://www.nature.com/articles/s41586-024-08328-6
> > >
> > > [LC-PFN] https://proceedings.neurips.cc/paper_files/paper/2023/hash/3f1a5e8bfcc3005724d246abe454c1e5-Abstract-Conference.html
> > >
> > > [ForecastPFN] https://proceedings.neurips.cc/paper_files/paper/2023/hash/0731f0e65559059eb9cd9d6f44ce2dd8-Abstract-Conference.html

---

### Decision · Program_Chairs · 2025-04-30

**Decision:**

Accept (poster)

**Comment:**

Excellent comments and discussions by the reviewers, but they all agree on acceptance.